# The System of Corrective Interventions in the Sex Offender Population and the Proposed “Trident Statal Program” (TSP) in the Field of Italian Sex Crimes

**DOI:** 10.3390/bs15081085

**Published:** 2025-08-11

**Authors:** Giulio Perrotta, Stefano Eleuteri, Simona Grilli, Giulio D’Urso, Irene Petruccelli

**Affiliations:** 1Department of Human and Social Sciences, Universitas Mercatorum, Piazza Mattei 10, 00186 Rome, Italy; stefano.eleuteri@unimercatorum.it (S.E.); simona.grilli@unimercatorum.it (S.G.); irene.petruccelli@unimercatorum.it (I.P.); 2Department of Law, Economics, and Human Sciences, Mediterranean University of Reggio Calabria, Via dell’Università 25, 89124 Reggio Calabria, Italy; giulio.durso@unirc.it

**Keywords:** sex offender, corrective interventions, psychotherapy, cognitive–behavioral therapy, multisystemic therapy, additional sex offender treatments, risk–need–responsivity, good lives, epidemiological criminology, trident statal program

## Abstract

The issue of effective treatments for individuals with a history of sexual offending has legal, social, economic, political, and clinical impacts. Studies conducted on the topic of evaluating treatment outcomes for sex offenders have examined both biological and psychological interventions. The etiology of the phenomenon appears to be multifactorial, as the sexual harasser learns from the social and family context the norms that will constitute our framework of rules, in addition to the characteristics of temperament, character, and personality. Therefore, there is a need for a definite social–health policy at the government level to be able to address the legal-judicial, socio-political, and health problem of Italian sex offenders to reeducate and reintegrate them into society by drastically reducing or nullifying the risk of recidivism. This study proposes the “Trident State Program” (TSP), which could reduce or solve the problem of recidivism of Italian sexual offenders and promote a better process of re-education and reintegration of these individuals. The operational protocol of the program is being drafted.

## 1. Sex Offenders: Definition; Psychosocial, Clinical, and Educational Contexts; And Risk of Recidivism

### 1.1. Introduction

Nonconsensual sexual behavior and the legal system that punishes these types of illicit behaviors represent issues of high social, political, and psychological impact in all civil systems that regulate a specific penal regime of a criminal nature ([37]; [42]; [52]).

Prevention, for more than half a century, has been the starting point in most of the literature, both in terms of first punishable conduct and recidivism ([41]), trying to explain through the system of interventions the best remedial approaches to the problem ([14]), and also because of the strong media impact of public debates on the issue of public and personal health ([76]; [63]).

However, the complexity of the analysis is determined by several factors, which must be considered and analyzed—the nature of the crime; the victimological assessment; the criminogenic profile and subjective characteristics; the criticality of the judicial system; and mistrust in the criminal justice system. All these variables make each sexual crime case unique and complex, making interventionist analysis a real social, legal, political, and public health challenge ([75]; [27]; [71]).

The criminal justice system of legal systems around the world distinguishes between different types of nonconsensual sexual behavior; we mainly speak of sexual abuse, sexual assault, and rape, according to an increasing scale of severity ([58]; [75]; [71]; [20]):(1)Sexual abuse consists of any kind of nonconsensual sexual contact, regardless of age and gender, social status, or any other interpersonal differences. It is physical and/or verbal harassment that is limited to the use of inappropriate verbal conduct with sexual content or inappropriate sexual behavior without the use of force or violence.(2)Sexual assault consists of sexual abuse that materializes physical conduct with sexualizing value, with the use of force or violence, excluding penetrative acts.(3)Rape is sexual violence characterized by using force or violence in the performance of one or more repeated penetrative acts, with the male sex organ or any other body part or object, even unconventional.

Socially, the problems associated with sexual crimes are numerous. First, different legal systems recognize the type of crime and its punishability differently, just as they define sexual assault differently from sexual violence. However, they often blur the boundaries of behavior, reducing the possibility of punishment and leaving the victim with the burden of providing proof of the harm suffered. Furthermore, these definitions may vary based on the provisions of individual national legal systems, which decide the severity and therefore the punishability ([75]).

Intervention measures for serial molesters around the world vary according to the severity of the crime (whether committed only once or repeatedly or systematically, whether committed alone or in a group, whether committed with violence and abuse of a dominant position or due to fortuitous circumstances that harm life, whether committed against a minor or an adult, or finally whether committed by a family member or a stranger), the type of crime (to be distinguished between violent forms of sexual harassment, verbal violence, physical violence, and with or without homicide), and the socio-environmental context (whether in an environment with or without social containment rules, whether in an economically poor or rich environment, and whether in a Western or Eastern context), but generally include clinical therapeutic interventions, legal approaches (such as legal restrictions, surveillance, and punishability of the crime), and, in some cases, control measures as occurs for the crime of stalking ([58]; [75]). Behavioral therapy, personality analysis, group therapies, psychiatric rehabilitation, and long-term monitoring are common approaches. Prevention, through education and awareness, is essential, as is collaboration between law enforcement and health institutions ([8]; [58]; [70]; [20]).

From an epidemiological point of view, statistics vary based on the main variables examined, such as the age of the victim and the perpetrator, the socio-environmental and geographical context, and the psychopathological characteristics of the subjects involved. It is not possible to estimate the percentage without referring to a specific group (for example, serial crimes committed by an adolescent or an adult, and minor or adult victims with specific biological and anthropometric characteristics) or a specific geographical region (for example, the United States of America, Northern Europe, Africa, or Japan), as the variability is so heterogeneous that it does not allow for a precise synthesis ([8]; [70]; [20]).

### 1.2. Social, Clinical, and Educational Contexts

In the literature over the past two decades, the numerous meta-analyses conducted on the topic of evaluating treatment outcomes for individuals with a history of sexual offending have examined both biological (pharmacological and integrative) and psychological (treatment) interventions, with good rates of recidivism reduction especially when combined, given the variables examined such as age, family, socioeconomic and school–work contexts, mental health outcomes and behavioral addictions, follow-up time, medium- and long-term treatment, level of awareness of one’s interpretation of reality, and the motivational process resulting from the re-education received ([73]).

The attempt to explain the phenomenon of sexual offending has always been the subject of research, trying to use different contexts of reference ([11]; [67]; [54]), as follows:(1)The “social-anthropological theories” (man as a social animal, patriarchy, social–familial context of reference, and control of territory for settlement) assume that man, being a social animal with selfish tendencies, is driven by a sense of dominance and possession, and therefore the use of force innately represents the highest expression of control to demonstrate his superiority.(2)The “psycho-educational theories” (behavioral learning, direct experience, and repetition of the act because of reinforcements) assume, on the other hand, that behavior is the consequence of the subjective, cognitive process of interpreting reality, and the repetition is derived from the reinforcements that, from time to time, have maintained or discouraged that conduct.(3)The “clinical theories” (the psychopathology and neurobiology of dysfunction) assume, finally, that behavior is the consequence of the subjective, cognitive process of interpreting reality, but that underlying it is a precise neurobiology of dysfunction arising from psychopathology.

Recent studies increasingly seem to favor the multifactorial hypothesis, a combination of all these theories that view the sex offender as the social product of his context, the human product of his clinical condition, and the evolutionary product of his education. This is because sex offenders form the norms of their internal system from their social and family contexts, while from their experiential context they learn the reinforcements that will determine their future behaviors; finally, from a clinical context, they learn the ability (or inability) to interpret reality, both functionally and dysfunctionally (in the case of sex offenders) ([55], [56], [57], [59], [60]; [61]).

### 1.3. Parameters of Intervention Success: Risk of Recidivism and Compensation

At the government level, the success of the intervention is mainly measured in terms of absence or reduction in the risk of recidivism of the offending conduct and reduction in the harm caused in terms of compensation ([4]; [79]).

Traditionally, the penalty under the criminal justice system has been imprisonment or a suspended prison sentence to be served in the community, combined with a treatment or rehabilitation program related to the nature of the offense and the risk of recidivism ([17]; [13]).

Rehabilitation, which is enshrined in the constitutional charters of all Western countries, is based on the principle that people must be put in a position to change their lives and make up for their mistakes, fulfilling the sanctioning demands of the relevant system but also putting themselves in a position to reintegrate into that specific society through a cognitive–behavioral framework that uses subjective shaping models on a proactive basis ([64]; [25]; [43]; [47]).

The last two decades have witnessed a structured attempt to de-pathologize sex offenders, preferring a more social–anthropological interpretation related to environmental contexts than to actual mental illnesses; in fact, the evidence emerges in the literature that inclusion of these individuals to rehabilitative treatment programs tends more toward a preference for intervention in terms of the risk of recidivism and comorbidities than on the nature of the offense itself and the personological profile ([62]; [33]; [49]).

This perspective could be methodologically incorrect, and perhaps for this reason the interventions used up to now that are present in the literature are not able to drastically reduce or eliminate the risk of recidivism of the crime. In the opinion of the authors, therefore, starting from the models used, it would be necessary to calibrate the new intervention hypothesis precisely on the person, evaluating first the personality profile, with both a structural and functional analysis, to study their specific functional and dysfunctional traits, and on the basis of that calibrate a detailed intervention ([60]; [50]; [23]).

Only after that is it possible to intervene on the basis of cognitive–behavioral profiles, adapting the intervention to the individual subject according to a precise organizational scheme aimed at the recovery, reintegration, and re-education of the subject ([60]; [61]).

In this context, the role of policy is central, as the success of the individual’s integration into the community may be reduced by the failure of the intervention and the social stigma resulting from the crime committed; in fact, by paradox, in an attempt to protect the community and the public, active policies in this area may increase risk rather than reduce it, if these are marked solely by an attempt to discourage recidivism by working on subjective desistance and resistance, or by working most of the time to foster restorative justice processes or the use of therapeutic communities that nurture an isolative rather than inclusive pattern ([69]; [21]; [31]; [45]).

Given the heterogeneity of individuals with sexual convictions, there are several individual factors that can act as potential moderators of treatment, including age, type of offense, level of risk, (non)completion of treatment, and level of coercion (i.e., whether treatment is mandatory or voluntary).

Studies in this area need more development and implementation before they can be considered effective in supporting the integrative schema, precisely because of the limitations of the studies themselves ([22]).

## 2. Methods, Aims, and Objectives

### 2.1. Methods

A narrative literature review of peer-reviewed articles on PubMed from January 1980 to December 2024, using the keyword “sex offender,” was conducted to identify and critically analyze research on correctional interventions in the sex offender population. A total of 788 articles were identified. The methodology of this review conformed to the PRISMA (Preferred Reporting Items for Systematic Reviews and Meta-Analysis) guidelines (Figure 1). The population of interest included adults over the age of 18 who were in prison or placed in detention in facilities suitable for rehabilitation and rehabilitation. Examples of interventions included psychological, clinical–pharmacological, educational, and legal–punitive therapies, provided by the facilities through the intervention of health care professionals and technicians. Only studies that had the greatest argumentative adherence to programmatic projects, whether structured or in the form of simple protocols, were selected, totaling 56 articles. This narrative review included research articles such as systematic reviews and meta-analyses, randomized clinical trials (RCTs), quasi-experimental studies (i.e., non-RCTs), observational studies, and pre-/post-intervention studies, totaling 81 articles. These articles described the importance of using a programmatic model in theory and the usefulness of implementing the protocol in practice to prevent or rehabilitate sex offenders from the risk of committing the crime or reoffending. Only publications in English with a high scientific impact were selected.

### 2.2. Aim and Objectives

The present editorial article aims to propose a new effective treatment program for sexual offenders (TSP: Trident State Program) through the dissemination of the theoretical model, which will be later structured into a protocol to be administered to a population sample for the pilot study. The primary objective is to investigate the topic of effective treatment for sexual offenders (with the acronym SOTP) in the literature, summarizing the general content of the topic and trying to identify strengths and weaknesses of the current most shared approach, while the secondary objective is to build a theoretical model that is potentially able to respond to the needs of innovation, adaptability, and applicability in reference to the Italian legal–economic context.

## 3. Singular and Multidisciplinary Approaches: Lights and Shadows in Comparison

In order to understand the etiology, process, treatment, and/or management of persons convicted of a sex offense, it is necessary to depart from the use of a single interpretive point of view, thus favoring a multidisciplinary approach that may be able to analyze all aspects of interest, starting from the analysis of the individuals involved to the assessment of the social–environmental context and individual and collective behaviors ([3]), with the intention of recovering, re-educating, and reintegrating the offender, essentially fostering a process of new psychophysical balance for the victim of the crime as well ([28]).

Such a holistic view would favor prevention as much as intervention, and would help subjective and collective integrative processes, provided that at the policy level there can be resources and projects intended for implementation in practice ([72]).

In the literature, it is not possible at present to favor a specific approach, albeit a multidisciplinary one, because existing approaches offer methodological challenges in many innovative interventions and community integration projects at the global level, but they are often organized for a small population sample and have a limited impact ([65]), in contrast to larger studies on treatment and risk assessment, where programs and working bodies are based on similar theoretical constructs ([44]).

From this it can be inferred that methodologically, large-scale, randomized, controlled trials may not be the most effective method for understanding individual change and effective risk management, while the best designs may be a combination of quantitative and qualitative research to provide a comprehensive view ([74]).

At issue, therefore, is not the analysis of the critical issues of the current models used (which have already been widely demonstrated to be effective) but rather the need to integrate them into a more systematic action framework that can promote multi-layered intervention (from the political to the social and clinical world), taking into custody the individual who has a criminal record for sexual crimes from the moment in which he is convicted to the moment in which he needs re-education and reintegration into society.

## 4. Individual and Collective Profile Management: The Principal Models Compared—A Methodological Discussion of the Introduction of a New Program

All active practitioners in the field, from therapists to social correctional policy officials to health care providers, are under increasing pressure to provide and foster the use of interventions that are effective, efficient, and cost-effective, according to an organizational rationale; therefore, it is essential to understand what the most durable and performance-based technical characteristics of individual interventions are according to a theoretical–practical model scheme ([35]).

The main evaluation factors relate to the program orientation and method of delivery, its content, its adaptability to the specific case, and the treatment setting. Meta-analyses in the literature are mainly oriented toward social–environmental approaches (such as criminological models) ([12]; [6]; [2]) and clinical approaches (cognitive–behavioral, group therapy, strategic-integrated, and psychodynamic) ([51]; [36]) and recombined with each other ([53]; [38]).

Since 2010, the Risk–Need–Responsivity model (RNR) has become the main model for penitentiary interventions at an international level. It has proven to be effective for various categories of offenders—women ([5]), people with mental disorders, people who are either extremely poor or have no financial problems ([1]), young people ([15]), and sexual offenders ([26]). The RNR model is the one that best responds to the principles of individualized and reintegrative intervention, and it seems possible to apply it to the Italian penal context ([80]).

In Table 1, the main approaches that are most used in clinical practice (in terms of psychiatric rehabilitation and re-education) are compared.

The most suggested models in the literature are based on the strengths of the treatment and rehabilitation of individuals with a prosocial purpose, according to the “Good Lives” Model (which represents the evolution of the “Risk-Need-Responsivity Model”), as rehabilitation is considered more of a process than an outcome by virtue of the cognitive–behavioral structure of the therapeutic intervention in which most programs/interventions for people convicted of sexual offenses are rooted ([25]; [51]).

Central to the critical analysis of the intervention are the offender’s cognitive and behavioral profile, and thus his mental health, but also his relationships with family and peers, occupation, education, lifestyle, and social–demographic factors ([19]; [66]); the model that tries to integrate all these elements is represented by epidemiological criminology (“EpiCrim”) ([32]; [77]), which seeks to respond to crime at the population level using public health approaches that work at the individual, interpersonal, community, and societal levels through all four stages of prevention (primary, secondary, tertiary, and quaternary) ([40]). This model emphasizes the importance of the individual, his or her relationships, and the broader social context, so that we can better understand his or her pathways to crime and develop rehabilitation programs that are fit for the intended purpose, reinforcing the thesis that sexual abuse is first and foremost a community problem, as well as an individual problem, and therefore requires an integrated, multidimensional response ([30]).

In Table 2, the main models that are the most effective are compared.

The following models, which are most widely used in clinical practice, reflect the State Health System’s attempt to curb the pathological phenomenon and remedy it through a structured, clinically based intervention.

However, despite this specialized effort, the deep-seated reasons for the failure of the full recovery of these are multifactorial, and depend on the degree of impairment of the patient’s clinical profile, the social–environmental context of reference and how it reinforces or does not reinforce the subjective psychopathological tendency, the work context and the subject’s ability to reintegrate into society and regain economic independence, and the support during follow-up by public and private agencies, organizations, and facilities dedicated to patients with this clinical problem.

The risk of failure and thus relapse is a problem that needs to be analyzed from a multifactorial perspective and needs to be addressed through a more in-depth analysis than just the clinical profile, which also and especially considers government intervention at the political and social levels.

There is an urgent need to find a specific program of intervention that can offer an organized, structured, and functional solution to the problem, one that starts from individual subjective needs but generally protects the fundamental rights of the community of the social context of reference.

## 5. The Complexity of the Current Corrective Action System and the Proposal of the “Trident Statal Program” (TSP) in the Field of Sex Crimes: Theoretical Framework and Structural Model

### 5.1. Foreword

Criminal justice policies, in all the world’s legal systems, are marked by a logic of the rationalization of resources, designed with respect to the costs and challenges of project implementation.

Public action must conform to criteria of efficiency, effectiveness, and the cost-effectiveness of intervention, and thus the results must be achieved with the least effort, to the maximum satisfaction expected prior to the intervention, and with the least financial outlay. This view, however, makes it difficult to apply interventions in all contexts in which the criminological–legal model must adapt to the needs of pedagogical–educational models and biopsychosocial models with re-educational purposes, as is the case in the hypothesis of individuals who commit sex crimes ([9]).

In such cases, government coverage of the service is provided by a public network through health facilities and contracted institutions, and a secondary care network of a private type, managed by nonprofit organizations and associations and private personal care services, which take charge of the subject, follows him from the early stages of rehabilitation to those of reintegration into society. This “binary” model of care has, over time, demonstrated various criticalities, fostering the dysfunctional dynamics noted in the literature, such as the risk of recidivism, partial or absent health care in the medium and long term after the prison sentence, and a social stigma of the criminal that persists even after re-education.

The proposed “Trident Statal Program” (TSP) aims to smooth out these shortcomings by fostering a system that can be functional in every concrete scenario related to sex offenders.

Table 3 contains a theoretical framework and model, structured into three specific interventions (or pillars).

### 5.2. “Primary Intervention”: The Preventive System and the Pedagogical–Educational Model

The first pillar structuring the program is devoted to the pedagogical–educational model of the preventive system, inspired by the German preventive intervention of the “Dunkelfeld Prevention Project” ([48]) and to the American model that established the figure of the consultant and mental health professional (school counselor and educational psychology) in a school context ([39]).

This intervention is based on the preventive principle of educational action and is dedicated to both the person who has not committed a sexual offense and the person who might commit it because he or she has a criminal inclination (both with a formative–educational function).

Specifically, this pillar involves the following actions:(1)Organization at the ministerial level of a specific educational policy (not modified) that provides for compulsory thematic meetings in middle–secondary schools to train the teaching staff and students about sexual offenses and defense tools.(2)Initiation of a psychological service desk in all school levels, regardless of grade, with the inclusion of at least one psychologist figure, framed by a national labor contract, with a permanent status, and with a managerial and organizational role, also supporting teaching and administration.(3)Centralized government management of policies to support the sex offender through participation in public calls for proposals with stringent legal requirements and making winning private organizations equal to public law entities, as supervised by the Ministry of Health.

### 5.3. “Secondary Intervention”: The Repressive System and the Criminological–Legal–Political Framework

The second pillar structuring the program is devoted to the criminological–legal model of the repressive system.

This intervention is based on the repressive principle of legal action and is dedicated to both the subject who has committed a sexual offense (with a punitive function) and the subject who might commit it because he or she is criminally inclined (with an anticipatory function).

Concretely, this pillar provides for the following actions:(1)Establishment of an inter-ministerial office dedicated to sex crimes, interfacing the Ministries of Education, Justice, and Public Security.(2)Tightening of the punishment regime for sex crimes, with repressive purposes and exclusion of rewards and sentence discounts.

### 5.4. “Tertiary Intervention”: The Clinical (Rehabilitation) System and the Biopsychosocial Model

The third pillar structuring the program is devoted to the biopsychosocial model of the rehabilitation system.

This intervention is based on the rehabilitative principle of psychosocial action and is entirely dedicated to the sex offender (with a reintegrative function).

Specifically, this pillar provides for the following actions:(1)Establishment of a personalized educational–training and clinical–psychological pathway, which is aimed at the re-education and reintegration of the sex offender already during the period of detention and throughout the subject’s life, with a follow-up function. The course must be educational in nature to enable the subject to find an independent personal and work placement in society, and it must be clinical in nature to offer him or her continuous personal support during and after the state of imprisonment and social reintegration.(2)Placement of the subject in the “Protection and Guardianship Program”, which gives a new personal identity to the sex offender and removes him or her from the social context of reference, to be reintegrated into a new neutral social–environmental context without, however, losing contact with close family members.

## 6. Methodological and Applicative Discussions

### 6.1. The Choice of the Model (Trident Statal Model, TSM)

The Trident model, in this area, represents a novelty in the legal–political and judicial–criminological landscape, as the prevalence is mono-interventionist or otherwise uncoordinated in multiple areas, as demonstrated in the literature during this review.

The pillars on which the entire program is built guide action on all sides involved in the model under analysis, which, due to its specific peculiarities, requires coordination between the legal, political–criminal, and clinical fields.

In fact, the failure of intervention for an individual is attributable to several variables. Subjective variables include age, gender, family background, personal history, subjective perception of one’s condition, and the severity of paraphilic symptoms, measured in terms of intensity and frequency. Objective variables include the economic and organizational conditions and the deviant–criminal outbursts of the social context, as well as working conditions, support from authorities and institutions during the sentence, rehabilitation, and social reintegration.

This new proposed model adapts to the multifactorial nature of the phenomenon, which intends to overcome this analysis asking too many questions, preferring instead a holistic approach to the problem by placing the sexual criminal at the center of the re-educational and rehabilitative process.

### 6.2. Model Description

To meet the needs of the program, the model is structured in three different pillars, which must coordinate to foster the result oriented precisely to the reduction or absence of recidivism of the crime and individual psychophysical well-being in relation to one’s clinical condition.

The pedagogical–educational model embodies the first pillar that performs preventive function with respect to the crime, with a formative–educational purpose, working on the process of subjective awareness.

In concrete terms, this pillar includes both actions at the government–ministerial level (such as the organization of a specific and non-modifiable educational policy at the local level, which includes mandatory thematic meetings in middle and high schools to train teaching staff and students on the topic of sexual crimes and on legal defense methods) and at the organizational–local level (such as the activation of a psychological help desk in all schools, regardless of the school year, with the inclusion of at least one psychologist with a management and organizational role who is also in support of the teaching and administration staff).

The aim is to extend the proposal to a transversal level, to ensure greater transparency and control by the bodies with decision-making power (such as the choice to ensure centralized management by the State in policies to support sexual crimes, through participation in public tenders, with stringent legal requirements and the equivalence of private entities awarded contracts to public law entities, and under the supervision of the Ministry of Health and/or Justice).

The centrality of decision-making power is essential to ensure decentralization, subject, however, to conditions of strict capillary control.

The criminological–legal model embodies the second pillar, which performs a repressive function with respect to the crime that has already taken place, either with an anticipatory purpose (in the sense of intimidating the potential subject about his or her intent via fear of punishment) or with a repressive–punitive purpose (in the sense of punishing the criminal and giving a strong and resistant signal to the community).

Specifically, this pillar includes actions at the government–state level, which are substantiated by the establishment of an inter-ministerial office dedicated to sexual offenses, which interfaces with the Ministries of Education, Justice, and Public Security, also by tightening the punitive regime for sexual offenses and excluding rewards and sentence discounts in cases of good behavior (the failure of which, during the time the sentence is served, will lead to specific aggravations of the sentence already imposed by the magistrate).

The importance of these interventions lies in the fact that repressing sexual misconduct at the punitive level is a deterrent to those who wish to commit such offenses, and thus a stricter regime could discourage such individuals to satisfy their illicit impulses.

The biopsychosocial model embodies the third pillar that performs a rehabilitative function with respect to the crime that has already taken place, with rehabilitative and reintegrative purposes in the sense of fostering both the re-education of the subject who is under the sanction regime and their reintegration into society in order to re-integrate him or her and allow him or her to return to live an adequate and dignified life, under conditions of respect and safety of his or her social community of reference.

Specifically, this pillar includes actions at both the individual and collective level, such as the establishment of a personalized educational–training and clinical–psychological path aimed at the re-education and reintegration of the sex offender during the period of detention and throughout the subject’s life, with a follow-up function, and the inclusion of the subject in the “Protection and Guardianship Program”, which gives him or her a new personal identity and removes him or her from the social context of reference in order to be reintegrated into a new, neutral social–environmental context without losing contact with close family members.

The purposes are clearly re-education and social reintegration, favoring the protectionist approach of the sex offender in comparison to the social stigma he or she receives for the serious crime committed.

Re-educating them sexually and reintegrating them into a protected and neutral context could foster the process of correcting the offending conduct, reducing the risk of recidivism.

The operating protocol that will be used for the first pilot study, scheduled for 2026, is currently being finalized and evaluated by the Ethics Committee consulted.

## 7. Limitations of the Model and Future Prospects

The proposed program represents a theoretical model that must be structured in its functional elements, to be administered first in a pilot study and then in a representative population sample of subjects who commit sexual crimes. The model is currently being approved by the Ethics Committee, who have been consulted to proceed with an exploratory investigation through a selection of a pilot population from a prison context. A fundamental limitation concerns the state-run nature of the program. Indeed, the program, in its entirety, must be submitted to a legislative process for finalization in a study sector with a representative population; therefore, the costs of establishment, management, and regulation would be borne by the state budget and, therefore, should be foreseen with specific economic–legal mention. Furthermore, depending on the nation (or the legal system of reference, whether civil law or common law), the model may already partially exist, and therefore it may not be necessary to define it ex novo but rather to remodel it based on the new program. Pending feedback from the Ethics Committee, the current limitations identified in making the protocol practical in the final drafting phase relate to the selection of the prison population sample (for the variables of age, type of crime, type of conviction, psychopathological profile, and pre- and post-crime conduct), to government authorizations to participate in the pilot study, and to the costs of carrying out the study; however, the challenge is stimulating, and the project will continue with the intention of making the model operational through the protocol.

## 8. Conclusions

Prison system policies toward the sex offender population is the foundation for implementing a structured and functional program that ensures a model inspired by the principles of prevention (in which the education system strengthens the concept of legality), repression (in which sexual offenses are severely punished), and rehabilitation (in which the offender is re-educated and reintegrated into society).

The current system integrates multiple models, depending on the context, but does not guarantee adequate positive coverage in terms of prevention and resolution of possible recidivism by sexual offenders themselves. The Trident State Program (TSP) aims to improve the current political, regulatory, and socio-criminological framework, in Italy and the Western world in general, to strengthen therapeutic intervention and its positive effects and reduce or eliminate the risk of sexual offense recidivism. Clearly, to design a program capable of best addressing and managing a complex and heterogeneous problem such as the management of sex crimes, an implementation phase is needed to test the protocol.

Future research will focus on completing the operational protocol for this model, obtaining government authorization to conduct a pilot study, and then a study with a representative sample of the prison population will be conducted to demonstrate the effectiveness, efficiency, and cost-effectiveness of the proposed new model.

## Figures and Tables

**Figure 1 behavsci-15-01085-f001:**
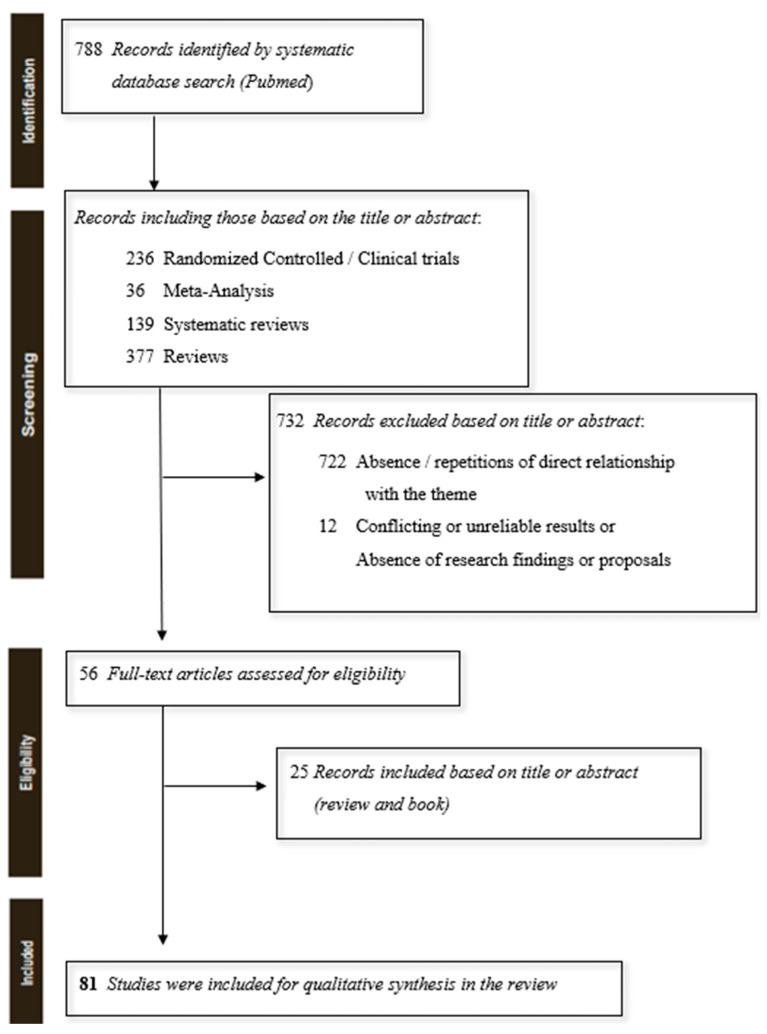
PRISMA flow diagram template for systematic reviews ([34]).

**Table 1 behavsci-15-01085-t001:** Comparison of the descriptions of the principal approaches used to plan rehabilitative interventions for sex offenders.

Approach	Description
Cognitive–Behavioral Therapy (CBT)	This approach is used in many therapeutic programs for sex offenders to prevent relapse. CBT combines two strategies to address both thoughts/conceptions and behaviors/actions, focusing work on one’s perceived (and thus cognitive focus), reinforcing patterns of behavior, while the behavioral component emphasizes actions that contribute to the act. This problem-centered approach helps sex offenders learn new skills and develop competencies to maintain appropriate behaviors through cognitive–behavioral deconstruction and restructuring work ([29]; [46]). Treatment programs involve various strategies that focus on correcting thoughts and emotions, regulating behaviors, and prosocial beliefs. CBT is also the most popular treatment approach for adolescent sex offenders and is widely supported ([10]).
Multisystemic Therapy (MST)	This approach has also been successful in treating sex offenders ([7]). Originally developed by Scott Henggeler as a family treatment program for antisocial children and serious offenders, it has evolved over time to work on adolescent socialization and interpersonal relationships, focusing on breaking the cycle of sexual violence to develop a safety plan ([18]). MST thus works on the systemic–relational aspects, and that is precisely why when combined with CBT more results are achieved ([68]).
Additional Sex Offender Treatments (ASOTs)	In addition to psychotherapies, sex offenders may also receive specific surgical (mechanical castration) and/or pharmacological (chemical castration) therapy. Specifically, the latter refers to hormonal drugs such as antiandrogens, which are used to reduce sexual arousal, although studies show that combination with psychotherapy is the best treatment choice, also considering the high rate of refusal or renunciation of drug therapies by patients ([24]). The paucity of studies related to mechanical castration leads to the inference that this practice is not conclusive ([16]).

**Table 2 behavsci-15-01085-t002:** Comparison of the strengths and weaknesses of the principal models used to plan rehabilitative interventions for sex offenders.

Model	Author	Strengths	Weaknesses
Risk–Need–Responsivity (RNR)	[2] ([2]); [5] ([5])	It was a widely used model of rehabilitation in the 1990s and 2000s, structured during the 1960s and 1970s, that focuses almost exclusively on risk reduction (recidivism) and the need to eliminate or reduce the consequences of harmful behaviors. There are three principles that should guide interventions: risk of recidivism, need to offend, and subjective reactivity. There is also an interaction with offender attributes; depending, for example, on age, gender, cognitive ability, or motivation, different types of interventions are indicated.	Failure to identify elements capable of motivating the subject to not reoffend.
Good Lives (GLM)	[78] ([78])	It was created to compensate for the shortcomings of the RNR and is presented as a rehabilitation model that focuses both on reducing the risk of recidivism and motivating the subject to stop reoffending sexually. The GLM, complementing RNR, provides a framework to help individuals build healthy, prosocial lifestyles aligned with their values and priorities, as opposed to the harmful behaviors for which they have been legally convicted. The Model accommodates the principles of risk, need, and responsivity, as it views a subject’s dynamic risk factors as signals of his or her lack of the ability to achieve so-called primary human goods (such as relationships, mastery, inner peace, and autonomy) in a prosocial manner. Dynamic risk factors are then addressed in the broader attempt to strengthen the client’s ability to achieve valuable goods through the acquisition of internal (skills and knowledge) and external resources (social supports, vocational training).	Motivation not to reoffend may depend on or be constantly reinforced by a dysfunctional or severe pathological personality structure, and therefore does not guarantee absolute certainty that the intervention will be successful in the long run.
Epidemiological crimino-logy (EpiCrim)	[77] ([77])	It integrates the previous models and seeks to respond to the issue of crime using a socio-environmental and clinical (public health and psychiatric) schema, which operates at both the individual/interpersonal and community/social levels through four stages of prevention (primary, secondary, tertiary, and quaternary). It emphasizes the importance of the individual, his or her relationships, and social context to better understand his or her pathways to crime and develop rehabilitation programs tailored to the person.	While working on both subjective and collective profiles, positive certainty of intervention is not guaranteed, as the individual motivational process comes into crisis at the time when the subject is the most vulnerable.

**Table 3 behavsci-15-01085-t003:** “Trident Statal Program” (TSP): the technical details of the model.

Pillar of Intervention	Area of Focus (System)	Type of Intervention	Description and Individual Proposed Actions
Primary intervention	Preventive system	Pedagogical–educational	This intervention is based on the preventive principle of educational action and is dedicated to both the person who has not committed the sexual offense and the person who might commit it because he or she has a criminal inclination (both with a formative–educational function); its actions are listed below: (1) Organization at the ministerial level of a specific educational policy (not modifiable); (2) Initiation of a psychological service desk in all school levels; (3) Centralized government management of policies to support the sex offender.
Secondary intervention	Repressive system	Criminological–legal–political	This intervention is based on the repressive principle of legal action, as follows: (1) Establishment of an inter-ministerial office dedicated to sex crimes, interfacing the Ministries of Education, Justice, and Public Security; (2) Tightening of the punishment regime for sex crimes, with repressive purposes and exclusion of rewards and sentence discounts.
Tertiary intervention	Clinical system	Biopsychosocial	This intervention is based on the rehabilitative principle of psychosocial action and is entirely dedicated to the sex offender (with reintegrative function), as follows: (1) Establishment of a personalized educational–training and clinical–psychological pathway; (2) Placement of the subject in the “Protection and Guardianship Program”.

Note. Intervention pillar: represents the variable related to the three different types of interventions covered by the program. Intervention area: represents the variable related to the three different types of areas covered by the program. Type of intervention: represents the variable related to the three different categories of intervention covered in the program. Description and proposed individual actions: represents the description of the intervention and the specific individual actions planned.

## Data Availability

No new data were created or analyzed in this study.

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
