# Peer review of "The System of Corrective Interventions in the Sex Offender Population and the Proposed “Trident Statal Program” (TSP) in the Field of Italian Sex Crimes"

_behavsci, 2025, doi:10.3390/bs15081085_

Round 1
Reviewer 1 Report
Comments and Suggestions for Authors
Very interesting and inspiring paper that discusses a very important topic.
As I am reading the paper without having a background in research that studies sex offenders I would be interested in hearing shortly more about where all these measures of intervention currently in use have their origins. Is there something more general that can be said about these measures globally, which ones are popular and where etc. Also, it would be good to hear shortly about sex offenders as a focus group, who they are, which kinds of statistics are maybe available.
Maybe also to crystallize more in the text, what exactly is the problem with sexual offending on the level of society and communities and why it would be so important to find more organic ways of dealing with it. Sexual assaults are often not recognized in the community and the boundaries between what is allowed and what is not can be blurred. Depending on who you are your right to self-determination might be belittled. Sexual assaults are also very serious form of violence that often have long-lasting consequences. They are hard to prove and leave the victim often with a heavy burden of showing the proof. These types of assaults also stigmatize both offenders and victims.
Maybe some thoughts also around what might go well and what might go wrong when the new approach is going to be piloted could be added in the conclusions. I also think that one cannot underline enough the importance of multilayered methods of intervention as often it occurs that we try to fix a multifaceted problem by only treating the acute visible symptoms, but never tackling the root cause.
There is a lot of contents in the brackets in the text, I would advise checking all these and remove the brackets always when possible.
Some small remarks:
Row 61 (SOTP) refers this to a sexual offenders treatment program?
Row 186 RNR model, and this is?
Row 187 not men?
Row 193 most used in clinical practice. In which context?
Author Response
Dear, Thank you so much for your invaluable contribution. We truly appreciate it. This topic is very important to us, as we are developing an active protocol with the Italian government, and this publication is the first step in the scientific process. In the introduction, I've included your requests regarding completion and epidemiology, and in the following section, I've further clarified the legal terms referring to the different types. Finally, in the conclusions, I've included two paragraphs, as you requested, to refer back to the work. SOTP is an acronym that indicates the totality of treatments currently used, and I've specified it in the text to make it even clearer, at your request, as I did with the acronym RNR. The feminine form is because it has proven particularly effective in that group, but it works in both. Finally, I've specified what I mean by clinical practice, in which contexts. I hope the changes will satisfy you. At your disposal. Giulio
Reviewer 2 Report
Comments and Suggestions for Authors
The manuscript overall provides a compelling proposal for addressing the complex issue of sex offender recidivism through the Trident Statal Program (TSP). The literature review, along with relevant references, provides a solid foundation. The authors effectively summarize existing models such as the RNR, GLM, and Epidemiological Criminology (EpiCrim), and highlight their strengths and limitations (Table 2). The TSP model also offers a clear structure into three specific pillars (Table 3) and is well-articulated.
Nevertheless, several main areas exist and are suggested for improvement:
First, the primary limitation of this work is the lack of empirical data to support the TSP’s efficacy, as this work is mainly a theoretical proposal. Although TSP has excellent potential in addressing the multifactorial nature of sexual offending, the authors should consider providing a more precise roadmap for empirical validation if possible.
Second, the 5. Methodological and Applicative Discussions could benefit from more clarity and specificity regarding the operationalization of the TSP, particularly the “Protection and Guardianship Program” (Page 11, line 391), which is insufficiently described. Also, this Program may raise some ethical concerns regarding feasibility, consent, and potential stigmatization in new communities. The authors should address those concerns more explicitly and clearly. In addition, the section reads descriptively, and specific steps may be beneficial for the readers.
Third, Section 6. Limitations of the Model and Future Prospects (Line 401-411) - is overly brief and should be strengthened. The authors should address empirical limitations, possible challenges, and any research concerns.
Fourth, 7. Conclusions (Line 412-424) is also brief and should adequately capture the main points from the manuscript and offer future research directions or suggestions.
Finally, the self-citation issue. The manuscript seems to include self-citations by an author (In total, seven citations from Perrotta G., 2019, 2020, 2021, 2022, 2023, 2024a, 2024b) (Line 471-498). The authors should be cautious not to over-rely on their work and should justify and use citations with care. Also, the list references should be reworked and formatted.
Hope the above comments help.
Thank you again for your time and effort.
Comments on the Quality of English LanguageSome language issues that I spotted. This is not a complete list. The authors are encouraged to proofread and edit their manuscript:
- Line 11: “… has a legal, social, economic, political, and clinical impact.” should be “has legal, social, economic, political, and clinical impacts.”
- Line 14: “… nature, in that the sex offender learns from the social and family context the norms…” The phrase "in that" is awkward and unclear.
- Line 29: "Nonconsensual sexual behavior and the legal system that punishes this type of illicit behavior represent issues of high social, political, and psychological impact..." The phrase "this type of illicit" is grammatically incorrect. The verb "represent" should be plural.
- Line 37: "The complexity of the analysis is determined by several factors, such as … and distrust of criminal justice, make each case..." The sentence is grammatically incorrect and hard to follow. Please rephrase the sentence.
- Line 47: "…..It is physical and/or verbal harassment that is limited to the use of inappropriate verbal conduct with sexual value or inappropriate sexual behavior….." The phrase "with sexual value" is awkward and unclear. “Inappropriate verbal conduct" and "inappropriate sexual behavior" read redundant, and the sentence is a bit wordy.
- Line 94: "from the social and family context we learn the norms that will form our rule box..." The use of "we" and "our rule box" is inconsistent with the formal academic tone. Who "we" refers to (authors, society, offenders, or anything else)?
- Line 330: "The failure of intervention on the individual subject is ascribable as much too subjective variables... as to objective variables..." The sentence reads wordily, and the phrase "ascribable as much too" contains a major grammatical error ("too" should be "to").
Author Response
Dear, I greatly appreciated your invaluable help and hope the changes you'll find will be satisfactory. Specifically, I've made a series of linguistic changes as requested, modified the abstract to reflect your changes, significantly expanded the introduction, explained the limitations of the theoretical model, and better focused on the usefulness of this publication prior to testing with the pilot population we're identifying while awaiting approval of the protocol, which is still being finalized. Finally, I've reworked the limitations and conclusions as requested, rewriting all the references with the necessary style. I hope everything is going well now. At your disposal. Giulio